# Exhausted Capacity of Bicarbonate Buffer in Renal Failure Diagnosed Using Point of Care Analyzer

**DOI:** 10.3390/diagnostics11020226

**Published:** 2021-02-03

**Authors:** Tomasz Gołębiowski, Mariusz Kusztal, Andrzej Konieczny, Magdalena Kuriata-Kordek, Ada Gawryś, Hanna Augustyniak-Bartosik, Krzysztof Letachowicz, Dorota Zielińska, Magdalena Wiśniewska, Magdalena Krajewska

**Affiliations:** 1Department of Nephrology and Transplantation Medicine, Wroclaw Medical University, 50-556 Wroclaw, Poland; mariusz.kusztal@umed.wroc.pl (M.K.); andrzej.konieczny@umed.wroc.pl (A.K.); magdalena.kuriata-kordek@umed.wroc.pl (M.K.-K.); adagawrys@gmail.com (A.G.); hanna.augustyniak-bartosik@umed.wroc.pl (H.A.-B.); krzysztof.letachowicz@umed.wroc.pl (K.L.); melirus@wp.pl (D.Z.); magdalena.krajewska@umed.wroc.pl (M.K.); 2Clinical Department of Nephrology, Transplantology and Internal Medicine, Pomeranian Medical University, 70-214 Szczecin, Poland; mwisniewska35@gmail.com

**Keywords:** metabolic acidosis, chronic kidney disease, bicarbonate

## Abstract

Background: Metabolic acidosis in patients with chronic kidney disease (CKD) is a common complication. A bicarbonate concentration in venous blood (V-HCO_3_^−^) is a key index for diagnosis and treatment initiation. The aim of our study is to evaluate usability of acid–base balance parameters of in blood taken simultaneously from peripheral artery and the vein. Methods: A total of 49 patients (median age 66 years [interquartile range IQR 45–75]), with CKD stage G4 or G5 were enrolled in this cross-sectional study. All patients were qualified for arteriovenous fistula creation in pre-dialysis period. The samples were taken during surgery, directly after dissection, and evaluated in a point of care testing analyzer. The arteriovenous difference in bicarbonate levels (Δ-HCO_3_^−^) was calculated. According to glomerular filtration rate (eGFR) the group was divided into Group A eGFR ≥ 10 mL/min/1.73 m^2^) and Group B eGFR < 10 mL/min/1.73 m^2^). Results: In Group A Δ-HCO_3_^−^ was significantly higher compared to Group B. No such differences were observed in the case of V-HCO_3_^−^. Δ-HCO_3_^−^ positively correlated with eGFR. The discriminative power of Δ-HCO_3_^−^ for predicting eGFR < 10 mL/min/1.73 m^2^ was 0.72 (95% confidence interval [CI] = 0.551–0.88; *p =* 0.01) which provided 67% sensitivity and 75% specificity. The best cut-off was 0.5 mmol/L. Conclusions: The Δ-HCO_3_^−^ lower than 0.5 mmol/L may be used as predictor of exhaust buffer capacity. The value of this tool should be tested in larger population.

## 1. Introduction

Metabolic acidosis (MA) is a common disorder in patients with chronic kidney disease (CKD). The major complications of the metabolic acidosis in CKD include increased muscle protein degradation, with muscle wasting [1], stimulation of inflammation, reduced albumin synthesis [2], hypertension [3], bone disease [4], progression of CKD [5], and an increase in mortality [4].

Venous blood is usually used to assess acid-base disorders in CKD patients [6,7]. Much more information can be drawn when both arterial and venous samples are taken at the same time and used for evaluation. Metabolic processes in peripheral tissues cause blood acidification in the capillaries, and therefore the consumption of bicarbonate for pH correction.

Naturally, simultaneous venous and arterial blood collection is technically difficult, may lead to complications, therefore such diagnostics is not a standard approach. However, in the case of patients qualified for fistula creation, it is possible to collect samples, directly after vessels dissection, during the procedure.

Most of the published studies have focused on the plasma bicarbonate level, a surrogate marker of the status of acid balance, and their impact on MA complications, glomerular filtration rate (eGFR) decline and survival. None has tested whether it is the optimal parameter for MA diagnosis, and more importantly, whether the venous bicarbonate level is the optimal indicator of the body’s acid balance, that can be used as a guide for MA correction. Data describing prevalence of MA indicate that even in advanced kidney disease normal bicarbonate levels (>22 mmol/L) are found in majority of patients [8]. It does not mean the absence of MA. The increased proton (H^+^) excretion, as a result of intensive renal aminogenesis, in patients with preserved residual renal function, is responsible for maintaining the equilibrium state. Despite a normal serum bicarbonate concentration, positive acid balance is observed and leads to acidification of muscle and the interstitial compartment of the kidneys [9]. This condition is called subclinical metabolic acidosis (SMA) [10]. Published data showed that administration of bicarbonate in MA may positively influences protein degradation [11], increases the strength of lower body muscles in CKD patients [12], and slows the decline of eGRF even in SMA [13].

The aim of our study is to assess arteriovenous gradient in acid-base balance parameters, in group of CKD patients, in stage G4 and G5. We paid particular attention to identify the strongest parameters that would indicate the need of starting dialysis treatment, due to the exhaustion of compensation buffers and find most reliable parameter for guiding correction treatment with bicarbonate.

## 2. Materials and Methods

The participants with CKD stage G4 and G5 were identified from all of the patients hospitalized to Department Nephrology between August 2019 and August 2020. 

The inclusion criteria for patients were as follows: (1) age 18 years and over, (2) chronic kidney disease in stage G4 or G5 qualified for arteriovenous fistula creation, (3) ability to provide informed consent (4).

The exclusion criteria included the following: (1) receiving dialysis, (2) receiving emergency inpatient care within 4 weeks, (3) malnutrition with albumin level lower than 3.2 g/d/L, (4) supplementation of bicarbonate or (5) not willing to participate. Written informed consent was obtained from each patient before entering the study. Of the 104 eligible participants, 43 were excluded due to already receiving dialysis, and 12 did not agree to take part in the study. Finally, 49 patients were recruited in the study.

Demographic and comorbidity data were collected from a direct interview and medical documentation. Comorbidity included cardiovascular disease (i.e., coronary heart disease and stroke), cancer and diabetes. Coronary heart disease was defined as self-reported complaints or on the basis of a history of myocardial infarction, coronary angioplasty or bay-pass grafting. Stroke included a history of transient ischemic attack in the past or an ischemic/hemorrhagic event with neurological consequences. Cancers comprised a history of previous neoplastic disease or an active disease. Diabetes patients were those who received insulin or oral antihyperglycemic agents.

During the standard procedure of AVF creation (done under local anesthesia, end of vein to side of artery, by experienced interventional nephrologist), 2 blood samples (both from the artery and the vein after their dissection) were taken for analytical tests using an analyzer (RADIOMETER ABL90 SERIES, Radiometer Ltd., Poland). This device is designed for point of care testing (POCT). The samples were taken at the same time and conditions, in order to reduce the laboratory error. Assessment of the difference in concentrations (AV gradient) (Δ = A − V) of the following parameters: pCO_2_, Carbon dioxide partial pressure; HCO_3_^−^, Bicarbonate; ABE, Actual Base Excess; SBE, Standard Base Excess; K^+^, Potassium; Na^+^, Sodium; Ca^2+^, Ionized calcium; Cl^−^, Chloride; AG, Anion Gap, AG (K^+^), adjusting Anion Gap for the contribution of serum potassium.

Glomerular filtration rate (eGFR), serum creatinine concentration (sCr) and urea were assessed from venous blood drawn as standard procedure from a peripheral vein. The enzymatic method was used for urea and the Jaffe’s method for creatinine evaluation. eGFR was calculated based on the MDRD formula [14,15]. The values of 2–3 measurements were taken during hospitalization related to the creation of vascular access and included a maximum period of two days before and two days after surgery. These values were averaged and expressed the current values. This procedure ensured the minimization of laboratory errors in the assessment of the kidney function.

Statistical analysis was performed using standard software (Statistica Version 13.3, StatSoft, Tulsa, OK, USA). Continuous variables between groups were expressed as median and interquartile range [IQR] and compared using the independent *t*-test or Mann–Whitney U test, based on the normality of the variables, tested using the Kolmogorov–Smirnov test. Categorical variables were expressed as absolute (*n*) and percentage (%) and compared using the χ^2^ test.

The group of patients was divided into two subgroups according to eGFR (Group A GFR ≥ 10 mL/min/1.73 m^2^, Group B GFR < 10 mL/min/1.73 m^2^). The level less than 10 mL/min/1.73 m^2^ (for the Group B) was selected as the threshold in accordance with the KDIGO guidelines, which point that this level is an indication to start renal replacement therapy [16].

Receiver operating characteristic curves were constructed to eGFR < 10 mL/min/1.73 m^2^ (Group B of patients) and evaluate the discriminant value of A-AG (Arterial Anion Gap), Anion Gap, AG (K^+^), adjusting Anion Gap for the contribution of serum potassium, Δ-pH (arteriovenous pH differences), Δ-HCO_3_^−^ (arteriovenous differences in bicarbonate), Δ-ABE (arteriovenous differences in Actual Base Excess) and Δ-SBE (arteriovenous differences in Standard Base Excess). Area under the curves (AUC), cut-offs, sensitivity, specificity, positive predictive value (PPV) and negative predictive value (NPV) were assessed.

The sample size for evaluation of the correlation between eGFR, sCr, urea and POCT parameters was decided based on following calculation. To achieve the significant level (i.e., type-I error rate) of 5% (α =  0.05) and the statistical power of 80% (β  =  0.2) we estimated the ideal sample size in study group to be 46. Taking into account the normal distribution of total eGFR, the Pearson correlation coefficient was adopted.

Ethics approval was granted by the Ethics Board of Wroclaw Medical University No KB-609/2019.

## 3. Results

### 3.1. Clinical Characteristic

Forty-nine patients (median age 66 years [IQR 45–75], 18 females (36.7%), with median eGFR 11 mL/min/1.73 m^2^ [IQR 9.5–14] were enrolled in this cross-sectional study.

Table 1 shows the clinical characteristics of patients with causes of renal failure and comorbidity, comparing groups A and B. Both groups were uniform with regard to the above-mentioned features. 

### 3.2. Comparison of Acide-Base Balance Parameters in Grups A and B

Based on the U Mann-Whitney’s test, the following differences were found between both subgroups: In group A significantly higher values of calcium (A-Ca^2+^), (V-Ca^2+^) in arterial and venous blood, lover anion gap (A-AG) and adjusting Arterial Anion Gap for the contribution of serum potassium A-AG (K^+^), in comparison to Group B was found. Arteriovenous diferences in bicarbonate and base excess (Δ-HCO_3_^−^; Δ-ABE; Δ-SBE) were higher in Group A with lower glomerular filtration rate than in group B (Table 2). As expected, the median eGFR was significantly higher in Group A than in Group B (12 mL/min/1.73 m^2^ [IQR11–15] vs. 8 mL/min/1.73 m^2^ [IQR 7–9.5]). Bicarbonate level below 22 mmol/L was found in 40 (82%) and 41 (84%) out of 49 patients, in blood samples taken from the artery and vein, respectively. There were no significant diferances in bicarbonate level in both gruops. However, median level of A-HCO_3_^−^ was not statistically significant higher in Group A of patients (19.9 mmol/L [IQR 18.3–20.7] vs. 18.3 mmol/L [IQR 15.4–21], *p* = 0.201) and similarlyV-HCO_3_^−^ was higher in Group A (17.4 mmol/L [IQR 14.95–20.1] vs. 16.8 mmol/L [IQR 15.5–20], *p* = 0.864). 

### 3.3. The Pearson Correlations of Urea, Serum Creatinine (sCr) and Glomerular Filtration Rate (eGFR) with Acide-Base Balance Parameters

The indicators of kidney function, i.e., serum urea, creatinine and calculated eGFR, significantly correlated with several acide-base balance parameters (Table 3). The strongest correlation was found for arteriovenous gradients in bicarbonate level (Δ-HCO_3_^−^), Actual Base Excess (ABE) and Standard Base Excess (SBE), Δ-pH, ArterialAnion Gap (A-AG) and adjusting Arterial Anion Gap for the contribution of serum potassium (A-AG (K^+^)). Figure 1 shows the relationship between eGFR and Δ-HCO_3_^−^. It is a positive correlation, indicating that with lower eGFR values, i.e., more advanced kidney disease, the arteriovenous diferances in bicarbonate concentrations decrease. No significant correlation were found between serum urea, creatinine and calculated eGFR with arterial bicarbonate (A-HCO_3_^−^) and venous bicarbonate (V-HCO_3_^−^).

### 3.4. Receiver Operating Characteristic (ROC) Curves for the Variables with Most Important Discriminative Power for Group B of Patients (eGFR < 10 mL/min/1.73 m^2^)

Six ROC curves were generated for the parameters that best correlated with the eGFR. On this basis, it was found that the highest discriminative power for GFR < 10 mL/min/1.73 m^2^ is Δ-HCO_3_^−^ with area under curve (AUC) 0.716. The cut point was determined at 0.5 mmol/L (Figure 2). For this point, the sensitivity, specificity equals 67% and 75%, respectively. 

## 4. Discussion

In the present study, we found that the difference in bicarbonate concentration, between the artery and the vein (Δ-HCO_3_^−^), decreases according to the progression of chronic kidney disease. This information was used for further consideration whether this parameter may indicate a very advanced uremic kidney disease, in which the possibilities for acidosis compensation are limited or even impossible. For this reason, patients were divided into 2 subgroups, in terms of glomerular filtration rate (eGFR), where the limit was a value of eGFR < 10 mL/min/1.73 m^2^. This cut-off point is recommended by KDIGO for initiating renal replacement therapy (RRT) [16]. Among the following six predictors: Δ-HCO_3_^−^, A-AG (Arterial Anion Gap), A-AG (K^+^), adjusting Arterial Anion Gap for the contribution of serum potassium, Δ-pH (arteriovenous pH differences), Δ-ABE (arteriovenous differences in Actual Base Excess), Δ-SBE (arteriovenous differences in Standard Base Excess) we have chosen Δ-HCO_3_^−^ as the best discriminator for eGFR < 10 mL/min/1.73 m^2^. Δ-HCO_3_^−^ of 0.5 mmol/L was proposed as the cut-off point in regard of the highest diagnostic value. In other words, an Δ-HCO_3_^−^ of less than 0.5 mmol/L is indicative for significant advancement of renal failure, the need for initiation of RRT and significant limitation for acidosis compensation by the body buffers. Other parameters, mentioned above and shown in Table 4, may be also used for this purpose, although their prediction power was lower. It should be noted that the concentration of bicarbonates from the vein and the artery, as separate indicators of the advancement of metabolic acidosis, did not significantly differ between groups A and B, however, we observed a week trend of lower bicarbonate values among patients with more advanced renal failure (see Results section). There was no significant correlation between serum bicarbonate with the current eGFR.

In general, the prevalence of metabolic acidosis among CKD patients is higher with more advanced kidney disease. Data from the NHANES III study, including 15.594 subjects, aged 20 years and older, indicate that the prevalence of overt MA, defined as a reduction of bicarbonate in venous blood serum below 22 mmol/L, was observed in 1.3%, 2.3% and 19.1% in CKD stage G3, G4 and G5, respectively [8]. In the MDRD Study recruiting 1.782 CKD patients, with eGFR 13–50 mL/min per 1.73 m^2^, there was a positive correlation between eGFR and serum bicarbonate concentration, and this relationship was significantly stronger at lower eGFR than at higher eGFR values [17]. Correspondingly, in a larger group of CKD patients, in stage G3-5, the serum bicarbonate level positively correlated with eGFR [18].

Despite we did not find a significant difference in bicarbonate concentration between Group A and B, probably due to small study group (N = 49), we have identified important markers of acid-base imbalance. Among them, following parameters, significantly correlated with the current kidney function (A-AG, A-AG (K^+^), Δ-ABE, Δ-SBE, Δ-pH, Δ-HCO_3_^−^, Δ-ABE, Δ-SBE—see description of abbreviation in Table 3). This is an additional justification that bicarbonate is not an appropriate parameter to assess acidosis disorders, due to complexity of physiological mechanisms that are responsible for its concentration in the serum. 

Metabolic processes generate volatile and nonvolatile acids. The amount of nonvolatile acids, produced by the body, during metabolism, is termed the endogenous acid production [10]. Volatile acid is expired through respiration as carbon dioxide, whereas nonvolatile acid (proton) is excreted by the kidney in the form of ammonium and titratable acid [19]. Organic anion salts that are found in plant foods are directly absorbed in the gastrointestinal tract and yield bicarbonate [10]. The difference between protons, derived from the metabolism of ingested protein, minus the difference between alkali, absorbed in the digestive tract (originating predominately from ingested fruits and vegetables), and organic acid anions lost in urine is the net production of endogenous acid net (NEAP). This whole amount of nonvolatile acid that needs to be excreted in order to maintain the daily acid-base balance. Approximately, 1 mEq of NEAP per kilogram of body weight occurs each day in adults [10] In the NephroTest Cohort study involving 1.065 patients in CKD stage 1–4, positive acid balance was detected in 31% of patients with stage 4 despite of normal bicarbonate level in 90% of participants [20]. In this study urinary ammonia excretion decreased with eGFR, whereas NEAP did not. Looking at whole spectrum of CKD patients, the prevalence of normal bicarbonate concentration (>22 mmol/L) seems to be a more frequent than lower (<22 mmol/L) [8]. However, this does not mean that the acid balance in the body is correct [10]. The state in which acid retention is present, in the absence of a decrease in serum bicarbonate concentration, is the subclinical metabolic acidosis (SMA) [10]. In those patients, proton retention takes place in the interstitial tissue and involves buffers other than the bicarbonate. In animal studies (in rats), a positive acid balance was observed after 2/3 nephrectomy, which leads to acidification of the interstitial compartment of the kidneys and muscles, despite the normal concentration of bicarbonate in the serum [9]. In addition, other factors may affect the bicarbonate concentration, including dietary intake of substances metabolized into acid, the rate at which these substances are absorbed through the gastrointestinal tract, medications, such as diuretics or phosphate binders, and variability in bone disease which may also affect the body’s buffering capacity [21]. 

This pathophysiological complexity of bicarbonate buffer and the fact, that in many patients, even in advanced CKD, the concentration of bicarbonate acidosis is normal, is probably the reason that it is so difficult to reach a positive correlation between eGFR and bicarbonate in small research. To achieve this effect, a much larger group of patients should be recruited for the study than in the current one. However, the authors’ intention was to compare several predictors of parameters that can be used for this purpose and arteriovenous bicarbonate gradient (Δ-HCO_3_^−^) seems to be independent of all these variables.

Metabolic acidosis, defined by the KDIGO recommendation, is recognized when serum bicarbonate is below 22 mmol/L and then requires pharmacological correction [16]. However, administration of bicarbonate at a concentration above 24 mmol/L, increases the risk of congestive heart failure, although there was no association with mortality or atherosclerotic events [22]. Furthermore, in a reanalysis of the Chronic Renal Insufficiency Cohort Study (CRIC Study), sustained serum bicarbonate concentration above 26 mmol/L was associated with increased risk for heart failure and death [23]. For this reason, the recommendations, regarding the acidosis correction, suggest optimum serum bicarbonate between 22 and 26 mmol/L, but this guideline is based on weak data. We think that the safety of such therapy should be based, on other indicators which more accurately reflect metabolic changes in the tissue. More precise indicators of exhaustion of the body’s buffer systems are needed. 

On the basis of the ROC curves, we have found that the cut-off value for Δ-HCO_3_^−^ below 0.5 mmol/L is one indicator, where the buffering capacity of the acids is critical. In these situations, we suggest starting acidosis correction, even if there are only mild features of metabolic acidosis, i.e., SMA, according to the above presented definition. However, this recommendation must be substantiated and confirmed in larger trials.

Our innovative method of assessing acid-base disorders should provide new light on the tissue mechanism of acidosis regulation in chronic kidney disease. We hope that in the future, simultaneous and direct blood sampling in point-of-care testing (POCT) device will precisely diagnose the patient requiring acidosis correction. Naturally, regular puncture of the artery is difficult to perform in CKD patients and is unlikely to be of routine clinical use. On the other hand, we propose a one-time test of both arterial and venous blood bicarbonate (with Δ-HCO_3_^−^ evaluation), which may facilitate the decision for beginning of bicarbonate supplementation. We think that further treatment control will be rather based on venous Δ-HCO_3_^−^, taking into account changes in venous Δ-HCO_3_^−^ concentration during therapy and extrapolation with baseline values evaluated during surgery. This, however, will require further research.

A few limitations should be underlined when assessing our findings. Owing to the cross- sectional study design, we were unable to comment on the dynamic nature of CKD progression and acid-base balance changes. Our data are also limited by the fact that we recruited CKD patients from only one unit with a low number of participants. However, we believe that the results should properly reflect the trend that other (beyond the serum bicarbonate) indicators of acidosis should be included in the decision making, regarding acidosis correction. 

## 5. Conclusions

Arteriovenous bicarbonate gradient (Δ-HCO_3_^−^) is the best discriminating factor of advanced kidney disease that identifies exhausting buffer capacity and is suggestive for exhausting buffer capacity. Introduction into clinical practice may change classical thinking and the clinical practice, i.e., guiding the correction of acidosis in a patient with CKD patients in stage G4 and G5. At present, however, the significance of this parameter in clinical practice requires further research, including a prospective evaluation on a larger number of patients with hard endpoints.

## Figures and Tables

**Figure 1 diagnostics-11-00226-f001:**
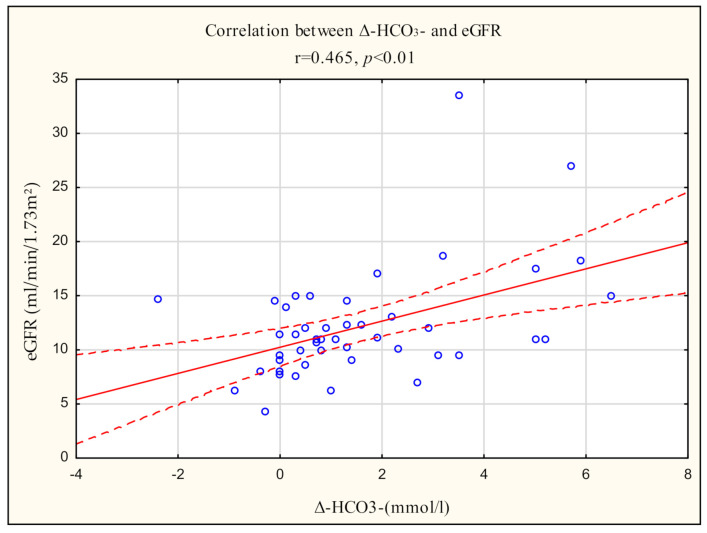
Pearson correlation between Δ-HCO_3_^−^ and eGFR in 49 CKD patients.

**Figure 2 diagnostics-11-00226-f002:**
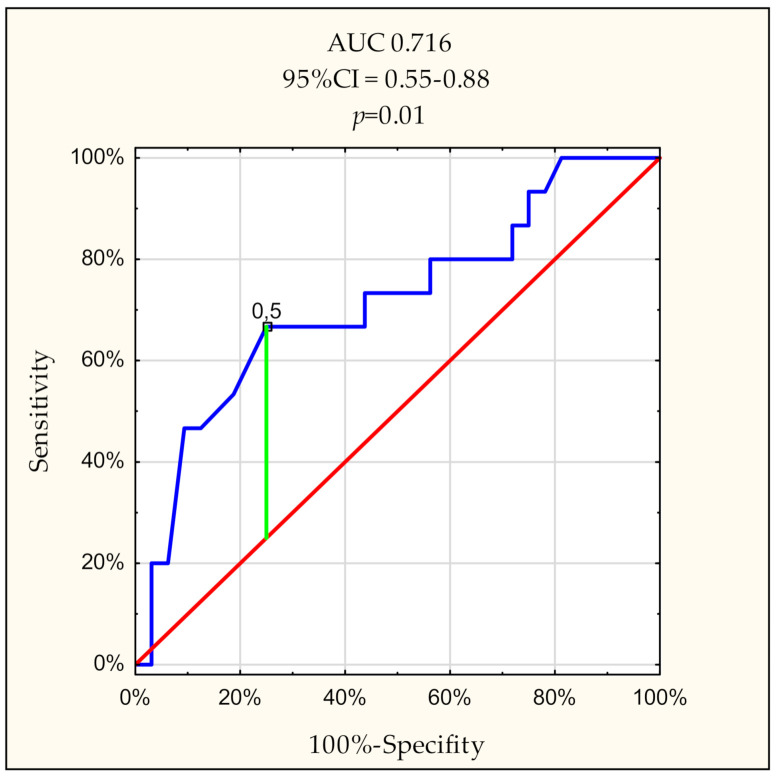
Receiver operating characteristic curve of Δ-HCO_3_^−^ on the prediction of GFR < 10 mL/min/1.73 m^2^ in 49 CKD patients. The proposed cut-off point was Δ-HCO_3_^−^ = 0.5 mmol/L.

**Table 1 diagnostics-11-00226-t001:** Baseline characteristics of the study group and comparison between Group A (GFR ≥ 10 mL/min/1.73 m^2^) and Group B (GFR < 10 mL/min/1.73 m^2^).

Variables	Group A GFR ≥ 10 mL/min/1.73 m^2^ N = 34		Group B GFR < 10 mL/min/1.73 m^2^ N = 15		
	Median	IQR	Median	IQR	*p*
Age (y.)	67	52–75	62	42–75	0.52 *
	Causes of CKD				
	No	%	No	%	
DM	13	38	8	53	0.82
Chronic GN	3	9	4	27	0.16
Others	18	53	3	20	0.15
	Comorbid conditions				
CHD	7	21	3	20	0.97
AF	4	12	1	7	0.62
PAD	6	18	3	20	0.87
Stroke	6	18	3	20	0.87
Malignancy	2	6	2	13	0.42
Present and past smoking	15	44	3	20	0.25

* student t test; χ^2^ test, *p* < 0.05 is considered statistically significant, comparing differences between Group A and B; CKD, chronic kidney disease; DM, diabetes mellitus; GN, glomerulonephritis; CHD, coronary heart disease, AF, atrial fibrillation; PAD, peripheral artery disease.

**Table 2 diagnostics-11-00226-t002:** Comparison between POCT and kidney function parameters in Group A and B.

Variables	Group A GFR ≥ 10 mL/min/1.73 m^2^ N = 34		Group B GFR < 10 mL/min/1.73 m^2^ N = 15		
	Median	IQR	Median	IQR	*p* *
A-pH	7.36	7.32–7.38	7.33	7.29–7.36	0.257
A-pCO_2_ (mmHg)	35.40	31.5–38	33.70	29.8–36	0.238
A-HCO_3_^−^(mmol/L)	19.90	18.3–20.7	18.30	15.4–21	0.201
A-ABE (mmol/L)	−5.40	−7.7–(−4.5)	−7.60	−11.3–(−4.1)	0.197
A-SBE (mmol/L)	−6.20	−8.50–(−4.9)	−8.40	−12.4–(−5)	0.197
A-K^+^ (mmol/L)	4.20	3.9–4.7	4.20	3.9–5.1	0.624
A-Na^+^ (mmol/L)	142.00	140–143	140.00	139–142	0.138
A-Ca^2+^ (mmol/L)	1.15	1.12–1.2	1.09	0.99–1.13	0.008
A-Cl^−^ (mmol/L)	112.00	110–116	113.00	107–117	0.647
A-AG (mmol/L)	9.50	8–10.6	10.60	10.1–11.9	0.023
A-AG (K^+^) (mmol/L)	13.80	13–14.8	15.60	13.9–16.4	0.020
V-pH	7.28	7.22–7.32	7.28	7.26–7.36	0.405
V-pCO_2_ (mmHg)	38.50	34.5–41.5	35.90	33.2–38.6	0.182
V-HCO_3_^−^(mmol/L)	17.40	14.95–20.1	16.80	15.5–20	0.864
V-ABE (mmol/L)	−8.80	−11.95–(−5)	−9.60	−11.1–(−5.1)	0.936
V-SBE (mmol/L)	−9.50	−12.75–(−5.3)	−10.50	−12.1–(−5.6)	0.991
V-K^+^ (mmol/L)	4.30	4.05–4.75	4.30	3.9–5.2	0.864
V-Na^+^ (mmol/L)	142.00	140.5–143	141.00	138–142	0.063
V-Ca^2+^ (mmol/L)	1.19	1.16–1.26	1.12	1.02–1.18	0.005
V-Cl^-^ (mmol/L)	114.00	109–117	110.00	107–116	0.247
V-AG (mmol/L)	11.10	9.45–12.9	11.70	10.5–13.2	0.288
V-AG (K^+^) (mmol/L)	15.75	14.25–17.25	16.60	16–17.5	0.222
Δ-pH	0.04	0.02–0.1	0.03	0.01–0.06	0.141
Δ-pCO_2_ (mmHg)	−1.85	−4.25–(−0.75)	−2.60	−3.70–(−1.6)	0.909
Δ-HCO_3_^−^(mmol/L)	1.30	0.55–3.05	0.30	0–1.4	0.019
Δ-ABE (mmol/L)	1.50	0.65–3.85	0.40	−0.3–(1.7)	0.024
Δ-SBE (mmol/L)	1.45	0.5–3.5	0.30	−0.4–(1.4)	0.024
Δ-K^+^ (mmol/L)	2.60	0–6.72	1.75	−2–(7.69)	0.366
Δ-Na^+^ (mmol/L)	0.35	0–0.72	0.70	0–0.71	0.653
Δ-Ca^2+^ (mmol/L)	−0.03	−0.07–(−0.01)	−0.01	−0.08–(0)	0.397
Δ-Cl^−^ (mmol/L)	0.00	−1–(1)	1.00	0–1	0.100
Δ-AG (mmol/L)	−1.55	−3.2–(−0.6)	−1.30	−2.1–(−0.1)	0.193
Δ-AG (K^+^) (mmol/L)	−1.85	−3.35–(−1)	−1.30	−1.9–(−0.1)	0.121
urea (mg/dl)	141.25	102.7–150.6	167.00	143–186.5	0.011
sCr (mg/dl)	4.33	4.12–5.15	7.47	5.05–8.35	0.000
eGFR (ml/min/1.73 m^2^)	12.00	11–15	8.00	7–9.5	0.000

* U Mann-Whitney’s (with continuity correction). Abbreviations: IQR, interquartile range; pCO_2_, Carbon dioxide partial pressure; HCO_3_^−^, Bicarbonate; ABE, Actual Base Excess; SBE, Standard Base Excess; K^+^, Potassium; Na^+^, Sodium; Ca^2+^, Ionized calcium; Cl^−^, Chloride; AG, Anion Gap, AG (K^+^), adjusting Anion Gap for the contribution of serum potassium.

**Table 3 diagnostics-11-00226-t003:** Pearson’s correlation of urea, serum creatinine (sCr) and eGFR with parameters of acid-base balance.

Variables	Urea (mg/dL)		sCr (mg/dL)		eGFR (mL/min/1.73 m^2^)	
	*r*	*p*	*r*	*p*	*r*	*p*
A-pH	−0.1238	0.412	0.0094	0.951	0.1588	0.292
A-pCO_2_ (mmHg)	−0.1119	0.459	−0.0876	0.562	0.0803	0.596
A-HCO_3_^−^ (mmol/L)	−0.1336	0.376	−0.0171	0.910	0.1481	0.326
A-ABE (mmol/L)	−0.1432	0.342	−0.0270	0.859	0.1557	0.301
A-SBE (mmol/L)	−0.1268	0.401	−0.0196	0.897	0.1406	0.351
A-K^+^ (mmol/L)	0.2930	0.048	0.0894	0.555	−0.0117	0.939
A-Na^+^ (mmol/L)	0.0359	0.813	−0.0642	0.672	−0.3073	0.038
A-Ca^2+^ (mmol/L)	−0.1347	0.372	−0.3166	0.032	0.3234	0.028
A-Cl^−^ (mmol/L)	−0.1134	0.453	−0.1680	0.264	−0.0704	0.642
A-AG (mmol/L)	0.4532	0.002	0.3506	0.017	−0.4357	0.002
A-AG (K^+^) (mmol/L)	0.5443	0.000	0.3811	0.009	−0.4444	0.002
V-pH	0.0388	0.798	0.2905	0.050	−0.1978	0.188
V-pCO_2_ (mmHg)	−0.0138	0.928	−0.1660	0.270	0.1527	0.311
V-HCO_3_^−^ (mmol/L)	0.0607	0.689	0.2171	0.147	−0.1385	0.359
V-ABE (mmol/L)	0.0515	0.734	0.2099	0.162	−0.1339	0.375
V-SBE (mmol/L)	0.0530	0.726	0.1954	0.193	−0.1183	0.434
V-K^+^ (mmol/L)	0.2137	0.154	0.0624	0.680	0.0013	0.993
V-Na^+^ (mmol/L)	0.0333	0.826	−0.1699	0.259	−0.1896	0.207
V-Ca^2+^ (mmol/L)	−0.2164	0.149	−0.3889	0.008	0.3820	0.009
V-Cl^−^ (mmol/L)	−0.1840	0.221	−0.2605	0.080	0.0289	0.849
V-AG (mmol/L)	0.3001	0.043	0.0951	0.530	−0.1513	0.315
V-AG (K^+^) (mmol/L)	0.3845	0.008	0.1184	0.433	−0.1610	0.285
Δ-pH	−0.1642	0.276	−0.3683	0.012	0.4028	0.006
Δ-pCO_2_ (mmHg)	−0.1426	0.344	0.1299	0.389	−0.1200	0.427
Δ-HCO_3_^−^ (mmol/L)	−0.3105	0.036	−0.3943	0.007	0.4647	0.001
Δ-ABE (mmol/L)	−0.3012	0.042	−0.3980	0.006	0.4604	0.001
Δ-SBE (mmol/L)	−0.3016	0.042	−0.3873	0.008	0.4426	0.002
Δ-K^+^ (mmol/L)	0.2021	0.178	0.0684	0.652	−0.0320	0.833
Δ-Na^+^ (mmol/L)	0.0129	0.932	0.1889	0.209	−0.2924	0.049
Δ-Ca^2+^ (mmol/L)	0.1862	0.215	0.1735	0.249	−0.1436	0.341
Δ-Cl^−^ (mmol/L)	0.1650	0.273	0.2059	0.170	−0.2620	0.079
Δ-AG (mmol/L)	0.1655	0.272	0.2933	0.048	−0.3246	0.028
Δ-AG (K^+^) (mmol/L)	0.1860	0.216	0.2958	0.046	−0.3198	0.030

Abbreviation: *r*, Pearson correlation coefficient; sCr; serum creatinine; eGFR, glomerular filtration rate; pCO_2_, Carbon dioxide partial pressure; HCO_3_^−^, Bicarbonate; ABE, Actual Base Excess; SBE, Standard Base Excess; K^+^, Potassium; Na^+^, Sodium; Ca^2+^, Ionized calcium; Cl^−^, Chloride; AG, Anion Gap, AG (K^+^), Anion Gap with the potassium cation included in the equation. *p*-value <0.05 statistically significant.

**Table 4 diagnostics-11-00226-t004:** Discriminative power of different acid-base indices for diagnosis eGFR < 10 mL/min/1.73 m^2^.

	AUC	95% CI	*p*	Proposed Cut-Off Point	Sensitivity (%)	Specificity (%)	PPV (%)	NPV (%)
A-AG	0.707	0.54–0.87	0.013	9.9 mmol/L	86.7	66.7	54.2	91.7
A-AG (K^+^)	0.712	0.55–0.88	0.012	14.3 mmol/L	73.3	66.7	50.0	84.6
Δ-pH	0.634	0.47–0.80	0.119	0.065	86.7	43.8	41.9	87.5
Δ-HCO_3_^−^	0.716	0.55–0.88	0.010	0.5 mmol/L	66.7	75.0	55.5	82.7
Δ-ABE	0.707	0.54–0.87	0.013	0.6 mmol/L	60.0	75.0	53.9	80.0
Δ-SBE	0.708	0.55–0.87	0.012	0.9 mmol/L	73.3	62.5	47.8	83.3

Abbreviations: AUC, area under curve, CI, confidence interval; PPV, positive predictive value; NPV, negative predictive value; A-AG, Arterial Anion Gap; AG (K^+^), adjusting Anion Gap for the contribution of serum potassium; Δ-pH, arteriovenous pH differences; Δ-HCO_3_^−^, arteriovenous differences in bicarbonate; Δ-ABE, arteriovenous differences in Actual Base Excess; Δ-SBE, arteriovenous differences in Standard Base Excess.

## Data Availability

The data presented in this study are available in this article.

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
