# Peer review of "Exhausted Capacity of Bicarbonate Buffer in Renal Failure Diagnosed Using Point of Care Analyzer"

_diagnostics, 2021, doi:10.3390/diagnostics11020226_

Round 1

Reviewer 1 Report

I have read with interest the manuscript by T. Golebiowski at al. They studied well known problem of metabolic acidosis in a group of 49 patients with chronic kidney disaese. The study as an attempt to answer a question if and when to start supplement biocarbonates in chronic kidney disaese patients. The method of blood smaple collection is unique, during a standart procedure of AVF creation therefore not exposing a patient to additional suffering. It produces a problem in everyday use of the method. Important point is a sample size. I belief it is a first step to further studies. The  study is well designed and performed. The manuscipt well written and stays according MDPI publisher guidlines.  

Author Response

Reviewer 1

I have read with interest the manuscript by T. Golebiowski at al. They studied well known problem of metabolic acidosis in a group of 49 patients with chronic kidney disease. The study as an attempt to answer a question if and when to start supplement bicarbonates in chronic kidney disease patients. The method of blood sample collection is unique, during a standard procedure of AVF creation therefore not exposing a patient to additional suffering. It produces a problem in everyday use of the method. Important point is a sample size. I belief it is a first step to further studies. The study is well designed and performed. The manuscript well written and stays according MDPI publisher guidelines.

Response: Thank you for the positive evaluation of our work.

Regular puncture of the artery is obviously difficult to perform in CKD patients and is unlikely to be used in daily clinical routine. On the other hand, we proposed a one-time test of both arterial and venous blood bicarbonate with Δ-HCO3- evaluation, which may facilitate the decision for starting bicarbonate supplementation. We believe that further treatment control will be rather based on venous HCO3- only, taking into account changes in venous HCO3- concentration during therapy and extrapolation with baseline values (made during surgery). This, however, will require further research.

We would like also to add that the sample’s volume is small but we are still working on it. Initially, we have qualified hemodialysis (HD) and non-dialysis patients for the study, and over 100 patients were included, however, during the preparation of the manuscript, taking into account the influence of bicarbonate supplementation during hemodialysis (from dialysis fluid), we decided to remove the HD group from further analysis. However, the results in this large group did not significantly differ from the results presented in our manuscript. For example: the discriminative power of Δ-HCO3-, for predicting eGFR<10ml/min/1.73m² was 0.66 (95% confidence interval [CI]=0.5-0.82; p=0.04). The best cut-off was 0.3mmol/l. (data not included in the manuscript).

Reviewer 2 Report

It is with interest that I reviewed this manuscript entitled "Exhausted capacity of bicarbonate buffer in 2 renal failure diagnosed using point of care analyzer". The Authors measured bicarbonate concentration in arterial and venous samples of advanced CKD patients undergoing arterio-venous fistula surgery and found that a the difference between venous and arterial bicarbonate was lost below 10 ml/min of eGFR. They found that the loss of arterio-venous bicarbonate delta predicted low eGFR.

I found that the paper is well written and data are meticulously collected and analyzed, but I do have some observations

1) Major observations

  • The main conclusion in this study is that delta HCO3 positively correlates with eGFR and low delta HCO3 predicts eGFR lower than 10 ml/min. However, as shown in table 3, with a r of 0.46, the correlation between eGFR and delta HCO3 is overall only moderate (with a strong p value). Also, I don't find particular clinical use in correlating arterio-venous HCO3 delta with eGFR, as creatinine levels are usually promptly available everywhere and delta calculation requires arterial blood sample collection, which, as we all know, is a more invasive procedure than usual venous blood sample (that can assess both eGFR and HCO3). I would suggest to find a more significant clinical correlation for this newly described biomarker, such as  nutritional status assessment, mean blood pressure values, etc
  • The "experimental section" should rather be "materials and methods" but it includes some data that should be moved into the "result" section (such as cohort descriptive data)

2) Minor observations

  • line 41 "the discriminative value of"...what? please clarify
  • line 51 MA is not a symptom but rather a sing
  • Line 73 would suggest to change "term" with "condition"
  • Line 127 "operational" should be changed to "operating"
  • Line 154 correct typo
  • Tables are overall difficult to read; I would suggest to change tables format including mean (or median) ± SD (or [range]); also, I don't understand why serum gas analysis data (pH, HCO3, etc) are listed as median followed by IQR rather than mean± SD. Please provide explanation
  • Line 163 correct typo (group)
  • Line 175 correct typo (creatinine)
  • Line 314 clarify/correct
  • Line 327 please clarify this statement
  • Line 343 "only (mild?) symptoms"

Author Response

Reviewer 2

It is with interest that I reviewed this manuscript entitled "Exhausted capacity of bicarbonate buffer in 2 renal failure diagnosed using point of care analyzer". The Authors measured bicarbonate concentration in arterial and venous samples of advanced CKD patients undergoing arterio-venous fistula surgery and found that the difference between venous and arterial bicarbonate was lost below 10 ml/min of eGFR. They found that the loss of arterio-venous bicarbonate delta predicted low eGFR.

I found that the paper is well written, and data are meticulously collected and analyzed, but I do have some observations

Response: We thank you very much for this positive opinion.

1) Major observations

The main conclusion in this study is that delta HCO3 positively correlates with eGFR and low delta HCO3 predicts eGFR lower than 10 ml/min. However, as shown in table 3, with a r of 0.46, the correlation between eGFR and delta HCO3 is overall only moderate (with a strong p value). Also, I don't find particular clinical use in correlating arterio-venous HCO3 delta with eGFR, as creatinine levels are usually promptly available everywhere and delta calculation requires arterial blood sample collection, which, as we all know, is a more invasive procedure than usual venous blood sample (that can assess both eGFR and HCO3). I would suggest to find a more significant clinical correlation for this newly described biomarker, such as nutritional status assessment, mean blood pressure values, etc

Response: Indeed, we have found the positive moderate correlation between eGFR and Δ-HCO3- and it was our primary aim. We have chosen GFR<10ml/min/1.73m2 as a threshold point beyond which renal compensation mechanisms are extremely impaired and lead to manifestation of uremia complications. Therefore, in this state of the disease, renal replacement therapy should be started in a short time, as recommended by KDIGO. These mechanisms apply also to the bicarbonate resources in the body, which are in a very limited range, i.e. there are no differences in their concentrations between artery and vein. Low Δ-HCO3- (<0.5mmol/l) may serve as a determinant of the limited buffer resources. We found this issue as important as the aforementioned correlation. We think also that this information is important from general pathophysiological point of view and can also be taken as an additional guide to start hemodialysis treatment.

We agree that regular puncture of the artery is difficult to perform in CKD patients and is unlikely to be of routine clinical use. On the other hand, we propose a one-time test of both arterial and venous blood bicarbonate (with Δ-HCO3- evaluation), which may facilitate the decision for beginning of bicarbonate supplementation. We think that further treatment control will be rather based on venous HCO3-, taking into account changes in venous HCO3- concentration during therapy and extrapolation with baseline values evaluated during surgery. This, however, will require further research.

The reviewer suggested that other clinical factors may correlate with Δ-HCO3-, and  we performed following correlation:

Body Mass Index BMI r=-0.3134, p=0.063,

albumin r=-0.0716, p=0.678,

parathormon PTH r=-0.2938, p=0.082,

Systolic Blood Pressure SBP r=-0.1817, p=0.289,

Diastolic Blood Pressure DBP r=-0.1232, p=0.474,

Mean Blood Pressure MBP r=-0.1636, p=0.340,

Natritional Risc Score 2002 (NRS2002) r=-0.0580, p=0.737.

All were not statistically significant (p>0.05) but perhaps an increase in the number of observations or other parameters will correlate more strongly with Δ-HCO3-.

The "experimental section" should rather be "materials and methods" but it includes some data that should be moved into the "result" section (such as cohort descriptive data)

Response: We thank you for this remark. The text has been changed accordingly.

2) Minor observations

line 41 "the discriminative value of"...what? please clarify

Response: We apologize for missing the word “Δ-HCO3-“.

line 51 MA is not a symptom but rather a sing

Response: We changed the symptom into a complication.

Line 73 would suggest to change "term" with "condition"

Response: Thank you for your suggestions. The text has been changed accordingly.

Line 127 "operational" should be changed to "operating"

Response: Thank you for your suggestions. The text has been changed accordingly.

Line 154 correct typo

Response: The text has been corrected.

Tables are overall difficult to read; I would suggest to change tables format including mean (or median) ± SD (or [range]); also, I don't understand why serum gas analysis data (pH, HCO3, etc) are listed as median followed by IQR rather than mean± SD. Please provide explanation

Response: The median is the main value reported in the manuscript. In Table 2 we show results from statistical analysis -  Mann-Whitney's U test, which in fact is median test (comparing medians between groups) and according to suggestion of our statistician, IQR in line with median better describes data distribution. It is generally known that the median is more resistant to outliers than mean value and it is more significant in smaller subgroups (No 34 and 15). Mean value was used only in the abstract for age due to the text volume limitation (200 words), however, to increase the clarity of the manuscript, all continuous variables are now presented as median and IQR. Tables have been also modified accordingly, to improve clarity.

Line 163 correct typo (group)

Response: The text has been corrected.

Line 175 correct typo (creatinine)

Response: The text has been corrected.

Line 314 clarify/correct

Response: We have changed the wholesentence into “However, this does not mean that the acid balance in the body is correct.”

Line 327 please clarify this statement”

Response: Bicarbonates assessed in the venous blood depend on many factors that were included in the discussion, therefore, in only large multicentre studies, the bicarbonate concentration correlated with the eGFR. We have changed this part to clarify it.

Line 343 "only (mild?) symptoms"

Response: We have changed this part into “mild features of metabolic acidosis”.

Round 2

Reviewer 2 Report

Thanks for the extensive editing.

I found this statement interesting and worthy being mentioned in the discussion

"We agree that regular puncture of the artery is difficult to perform in CKD patients and is unlikely to be of routine clinical use. On the other hand, we propose a one-time test of both arterial and venous blood bicarbonate (with Δ-HCO3- evaluation), which may facilitate the decision for beginning of bicarbonate supplementation. We think that further treatment control will be rather based on venous HCO3-, taking into account changes in venous HCO3- concentration during therapy and extrapolation with baseline values evaluated during surgery. This, however, will require further research."

Author Response

Reviewer 2

I found this statement interesting and worthy being mentioned in the discussion

 "We agree that regular puncture of the artery is difficult to perform in CKD patients and is unlikely to be of routine clinical use. On the other hand, we propose a one-time test of both arterial and venous blood bicarbonate (with Δ-HCO3- evaluation), which may facilitate the decision for beginning of bicarbonate supplementation. We think that further treatment control will be rather based on venous HCO3-, taking into account changes in venous HCO3- concentration during therapy and extrapolation with baseline values evaluated during surgery. This, however, will require further research."

Response: Thanks again for the effort put into reviewing the manuscript. Naturally, the above-mentioned text has been introduced in the discussion section.